# PMK—A Knowledge Processing Framework for Autonomous Robotics Perception and Manipulation

**DOI:** 10.3390/s19051166

**Published:** 2019-03-07

**Authors:** Mohammed Diab, Aliakbar Akbari, Muhayy Ud Din, Jan Rosell

**Affiliations:** Institute of Industrial and Control Engineering, Universitat Politècnica de Catalunya, 08034 Barcelona, Spain; aliakbar.akbari@upc.edu (A.A.); muhayyuddin.gillani@upc.edu (M.U.D.)

**Keywords:** perception knowledge, knowledge-based reasoning, task-motion planning

## Abstract

Autonomous indoor service robots are supposed to accomplish tasks, like *serve a cup*, which involve manipulation actions. Particularly, for complex manipulation tasks which are subject to geometric constraints, spatial information and a rich semantic knowledge about objects, types, and functionality are required, together with the way in which these objects can be manipulated. In this line, this paper presents an ontological-based reasoning framework called Perception and Manipulation Knowledge (PMK) that includes: (1) the modeling of the environment in a standardized way to provide common vocabularies for information exchange in human-robot or robot-robot collaboration, (2) a sensory module to perceive the objects in the environment and assert the ontological knowledge, (3) an evaluation-based analysis of the situation of the objects in the environment, in order to enhance the planning of manipulation tasks. The paper describes the concepts and the implementation of PMK, and presents an example demonstrating the range of information the framework can provide for autonomous robots.

## 1. Introduction

The increasing complexity in robotic systems and in manipulation tasks requires sophisticated planning mechanisms to plan in a human-like way. Some classical task planning approaches, such as Fast Forward (FF) [1], use the Planning Domain Definition Language (PDDL) [2] to describe the world. Although this way of description is a common approach, it makes the closed world assumption, i.e., if some facts about the world are not known or change, a planner may not be able to find a solution. This limitation means that robots are not able to begin a task until all objects in the environment are known and the actions the robot can do on them are completely defined. To tackle this issue, alternatively, knowledge-based planning approaches like [3] have emerged as a new domain of planning, focused on making the robot able to perform complex manipulation tasks. These capabilities include the capture of a rich semantic description of the scene, knowledge about the physical behavior of the objects, and reasoning about the potential manipulation actions. However, in complex manipulation problems that require many task constrains such as the Towers of Hanoi problem or those presented in [4] where task and motion levels are coupled, it could be a challenge to compute a long sequence of actions with feasible motion solutions.

The knowledge-based reasoning approaches for planning like [5] are proposed to facilitate the process of manipulation by adaptively providing the required components for planning. These approaches have the capability to make the world open to cope with environmental dynamic entities. They can be used for complex manipulation tasks that require the combination of task and motion planning (TAMP) and the integration with a perception module like [6] can provide a rich semantic description for the robot whenever needed. In this sense, the well-modeled knowledge representation plays a significant role.

Many ways are used for knowledge representation, such as ontologies, that are concerned with structuring concepts and relations such that they are usable for reasoning tasks done by artificial systems (e.g., robots). Formally, an ontology is defined as “an explicit, formal specification of a shared conceptualization” [7]. The conceptualization refers to the abstract models of entities in a certain domain. These models are achieved by defining their relevant concepts along with their relations.

Some knowledge-oriented approaches, such as [6], used terminologies and inference mechanisms (way of querying and reasoning over the knowledge) to facilitate the planning process. Others, like [3], proposed action description models for the manipulation domain. However, on the one hand, these approaches are more task-specific than generic frameworks. Therefore, the need for a generic, well-defined, and well-organized knowledge representation with common vocabularies has emerged to broaden the scope and to facilitate heterogeneous robot collaborative tasks. In this sense, some standardized ways of representing knowledge have been proposed, such as CORA [8], to provide a set of generic robotic concepts in order to share a common vocabulary and avoid ambiguities.

On the other hand, to integrate knowledge in TAMP approaches, some missing components are required such as action feasibility, placement regions, reachability, and manipulation constraints. These components play a significant role for bi-manual robot tasks or for multi-robot cooperation. For instance, it could be useful in planners such as [9] that require some geometric reasoning, or in physics-based planners like [10] that require to query about how interaction with the objects is to be done. Also, a knowledge-based sensing module could be integrated with TAMP contingency planners coping with uncertain scenarios, that require reasoning about the sensing system, the features of the environment entities, and sensor limitations. This way, the robot could be aware about the type of sensors that it has and how to use them.

This paper contributes covering the shortages mentioned by proposing an open-source ontology-based knowledge representation framework, called Perception and Manipulation Knowledge (PMK), that includes some reasoning processes for autonomous robots to enhance Task and Motion Planning (TAMP) capabilities in the manipulation domain. PMK has been modeled by adapting the available concepts provided by IEEE-1872 standards [11] of knowledge representation for the robotic domain. Moreover, some uncovered concepts related to manipulation domains have been proposed, such as knowledge related to sensors. This portable way of representing these concepts and relations is used to semantically link low-level perception data with high-level knowledge, and to analyze the situation of the environment entities in order to enhance the planning process for manipulation table-top problems. Furthermore, most of the required components for TAMP are introduced. The main contributions are, therefore, related to knowledge formulation and reasoning:Knowledge formulation includes:*(1) Standardized framework:* the formalization of a previously introduced ontology framework [12] in such a way that it follows the standardized concepts of representing the knowledge for the autonomous robotics domain.*(2) Knowledge representation for perception:* the extension of the previous framework [12] to include the perceived information from sensors. Particularly, a sensing class has been added to define sensors, measurements processes, and their relation with the robot, e.g., the representation is workable for cameras or Radio Frequency Identification (RFID), and may include any implemented sensing library such as Yolo.Reasoning includes:*(1) Situation analysis:* the development of inference process predicates based on Description Logic (DL) to evaluate the objects’ situation in the environment based on spatial reasoning, and to relate the classes entities and reason over them. Moreover, potential placement region and spatial reachability of the robot are introduced.*(2) Planning enhancement:* the use of PMK as a black-box for any planner to reason about TAMP inference requirements, such as robot capabilities, action constraints, action feasibility, and manipulation behaviors.

A motivation example is included that introduces a manipulation table-top problem at IOC-Lab, as shown in Figure 1. Two robots, the dual-arm YuMi robot and the TIAGo mobile manipulator, share the knowledge between each other in order to perform the task of serving beverages on a table. The set-up is prepared by YuMi, while TIAGo is used to actually serve the beverages to the customers.

This paper is organized as follows. Section 2 describes some related work, while Section 3 presents the system overview and the structure of PMK. Section 4 presents the PMK framework formulation, and Section 5 shows the case study. Section 6 presents an overview discussion of the results comparing with other approaches. Finally, Section 7 presents the conclusions of the work.

## 2. Related Work

### 2.1. The Use of Knowledge in Different Domains

Many studies have investigated the use of knowledge in planning, like [13,14], that categorize knowledge about the world into terminological knowledge (TBOX), and assertional knowledge (ABOX). The former contains a hierarchy of concepts, such as interaction and action, and their relations, whereas the latter contains individuals that are instantiations of these concepts. In the navigation area, some works such as [6,15] use a metric map and a topological map to define the robot environment. The metric map is used for the geometrical representation of the robot workspace in terms of free and occupied areas, while the topological map is used to capture the topology of the workspace. In the manipulation planning domain, works such as [12] propose an ontological framework to organize the knowledge needed for physics-based manipulation planning, allowing to derive manipulation regions and behaviors, or [16] that proposes ontologies to separately describe the knowledge on the manipulation objects and on the manipulation actions.

### 2.2. Standardization Efforts

The working group on ontologies for robotics and automation (ORA WG), sponsored by the IEEE Robotics & Automation Society, proposed standardized ways of representing knowledge for service, industrial, and autonomous robots [17]. ORA WG has proposed the Core Ontology for Robotics and Automation (CORA) [8], a conceptual structure to be used and integrated within other specific ontologies developed for the robotics and automation domain, i.e., with a main focus on reusability.

The structure of CORA is based on the Suggested Upper Merged Ontology (SUMO) [18], which is a top-level ontology that aims to define the main ontological categories describing the world. Later, in order to provide the more generalized concepts of the robotic domain, CORA was extended by incorporating CORAX [19], RPARTS (Robot parts), and POS (Position) [11]. CORAX is an ontology that defines the concepts not explicitly or completely covered by SUMO and CORA, such as robot motion (that categorizes the motion of the robot according to its type such as robot rolling or walking). RPARTS provides a general information of robot parts, also defining the attached parts on the robot such as robot sensing parts. POS captures general information about position and orientation. More recently, CORA has been enhanced by adding the knowledge, called Autonomous Robot Architecture Ontology (ROA), which defines the main concepts and relations regarding robot architecture for autonomous robots [20]. The IEEE 1872 standard covers all these ontologies and provides them in OWL form (https://github.com/srfiorini/IEEE1872-owl).

Many robotic tasks in real environments need knowledge for general task, motion, and manipulation planning. However, on the one hand, many knowledge frameworks have been presented that are not generic frameworks but task-specific. On the other hand, standards like CORA provide a generic knowledge, but with some concepts not fully covered, such as knowledge related to sensors, contexts in terms of spatial and temporal relations, and task representation. This paper contributes to fill this gap and proposes a standardized (unified) ontological framework (PMK framework) that covers from low-level robot data about perception to high-level environment information for manipulation purposes. The concepts and relations have been modeled and extended according to the IEEE 1872 standard.

### 2.3. Task and Motion Planning Inference Requirements

Task and motion planning combines the discrete action selection of task planning with the path generation by motion planning. For example, Figure 2 abstractly shows some significant necessary requirements that can be provided by the knowledge to the task and motion planning levels. Some questions arise at the task planning level such as: *does a given precondition hold?*, *how should feasibility checks of the actions be performed using semantic attachments? [21]*, *what is the spatial situation of the objects?*. Others arise at the motion planning level such as: *how should objects be manipulated?*, *how should the motion feasibility be checked?*. Many TAMP approaches need to reason about some necessary requirements in different robotics applications. For instance, knowledge-based TAMP has been proposed to solve the navigation among movable obstacles problem (NAMO) of a multi-robot system [5], where reasoning is done on the feasibility of the push and pull actions needed to move the obstacles away to clear the path towards the goal. A knowledge-based TAMP enhanced with a cloud engine has been proposed in [22], where the knowledge stores robots’ expertise on manipulation and navigation tasks and the cloud engine provides tools to efficiently retrieve this knowledge to analyze the requirements of new task and motion planning problems. A heuristic-based task and motion planning approach is proposed in [9] to solve table-top manipulation problems for bi-manual robots, where a relaxed geometric reasoning (regarding for instance reachability issues) is addressed in the computation of the heuristic. A constraint-based approach called Iteratively Deepened Task and Motion Planning, IDTAMP [23] iteratively generates candidate task plans, checks on their feasibility, and reasons on the failures to further constrain the search.

From the aforementioned approaches, a list of requirements to reason about in TAMP problems can be made: (1) the appropriate interaction parameters (such as friction, slip, or maximum force) required by physics-based motion planners to correctly interact with rigid bodies; (2) the spatial relations (such as *on, inside, right*, and *left*) between the objects on a cluttered scenario; (3) the feasibility of actions due to geometric issues like arm reachability, collisions, or the availability of object placements; (4) the feasibility of actions due to the object features (e.g., overweight objects out of robot capability); (5) the geometric constrains that limit the motion planner space [24]; (6) the action constraints regarding the interaction with the objects (e.g., a *cup* is graspable from the handle and pushable from body); and (7) the initial scene for the planner regarding for instance the potential grasp poses.

PMK provides knowledge and reasoning for these kind of requirements. Moreover, some other semantic information is also provided as a complement to the geometric feasibility of actions (e.g., an action can be geometrically feasible, although it may not be the type of action a human would logically do, like to place a cup on top of a mobile). PMK also aims to include this type of semantic reasoning.

## 3. System Formulation

### 3.1. System Overview

A robotic system is proposed, shown in Figure 3, which is composed of: a perception module, the PMK framework, a TAMP planning module, and an execution module. In the perception module, the tags are used to detect the poses and IDs of world entities and asserting them to the PMK to build the IOC-Lab knowledge. The PMK framework with reasoning mechanisms is used to provide the reasoning predicates related to perception, object features, situation analysis, and geometric reasoning.

These reasoning components, discussed in detail in Section 3.3, facilitate the planning process. The TAMP module is a combination of the FF task planner and physics-based motion planning [25]. It is used to actually plan the task and provide a feasible sequence of actions to the robot to be executed. The execution module uses YuMi and TIAGo robots to execute the serving task.

### 3.2. PMK Knowledge Structure

A preliminary version of PMK structure has been presented in [12]. It was inspired from an ontological schema proposed in [6] for navigation tasks in indoor environments, that describes concepts through the use a hierarchy of ontologies composed of three layers: metaontology, ontology schema, and ontology instance as shown in Figure 4. Metaontology is used to represent generic information, such as the concept of physical object. Ontology schema is used for domain specific knowledge, for instance, knowledge related to kitchen environments. Finally, ontology instance is used to store the information of the particular objects, such as a given cup and its features. Because an ontology is an object-oriented and frame-based language, the metaontology layer can provide a template for the ontology schema layer, while the ontology instance layer can be defined as an individual frame. The information of ontological classes, properties, and instances is transferred with bidirectional reasoning in the same knowledge layer, whereas unidirectional reasoning relates several knowledge classes of different layers.

Following this hierarchical schema, we proposed PMK framework (https://sir.upc.es/projects/ontologies/) for automated manipulation tasks where these layers are composed of seven classes: *Feature, WSobject, Actor, Sensor, Workspace, Context Reasoning,* and *Action* (Some concepts’ names have been modified here with respect to how they were presented in [12], in order to fit with the current standardized proposal). Each of them have three gradual levels (these levels will be discusses in Section 4). *Feature class* represents the knowledge related to the features of the objects such as physical interaction parameters and perceptual features, *WSobject class* is the knowledge related to the workspace objects and their components, *Actor class* is the knowledge related to the robots and their components, *Sensor class* is the knowledge related to the robot sensors or to sensors fixed in the environment, *WSpace class* is the knowledge related the workspace, *Context Reasoning class* is the knowledge related to the situation based on space and time, and *Action class* is the knowledge related to the planning processes including motion, perceptual, and manipulation components.

A preliminary version of this framework basically provided the concepts, relations, and inference mechanism for physics-based motion planning to teach the robot how to interact with the rigid bodies. However, the information regarding sensing was not considered, concepts that were proposed were not standardized concepts, and no knowledge regarding spatial and geometric reasoning for combined task and motion planning requirements was included. The PMK framework proposed here covers all the above missing issues.

### 3.3. PMK Reasoning Mechanism

As discussed in Section 2.3, the required inference for TAMP consists of: geometric reasoning to determine robot reachability and placement region, manipulation constraints analysis to determine how to interact with the objects, and the motion analysis to determine action feasibility.

The reasoning mechanism of PMK is divided into four parts, as shown in Figure 3: reasoning for perception, reasoning for object features, reasoning for situation, and reasoning for planning. Reasoning for perception is related to sensors and algorithms to simply answer questions like *which are the sensors the robot has? what is the corresponding algorithm to extract the perceptual data from the sensor?*. Reasoning for object features copes with the features of the objects such as color and dimensions. Reasoning for situation analysis is used to spatially evaluate the objects relations between each other (i.e., cup *inside* box, and *cup* is reachable by *left arm*). Reasoning for planning is used to reason about the preconditions of actions, action constraints, and geometric reasoning (arm reachability, grasping pose reachability, and placement region).

### 3.4. Why PMK?

This section covers the differences between PMK and other knowledge-based processing frameworks, such as KnowRob [3] and OUR-K [6], by highlighting the importance of PMK that is not covered by them. We divide the differences into two levels: modeling and reasoning process. At the modeling level, although these approaches provide frameworks that include a comprehensive way of representing the ontologies related to how to execute manipulation tasks, they lack the representation of the meta-level concepts using the common vocabularies provided by standardization such as [8,18] (i.e., they do not follow any standardization). This may lead to difficulties in incorporating/importing other ontologies under their terminologies because of the conflict in the meaning of the concepts for the robot. This may be a problem for collaborative tasks between robots that require some common vocabularies.

At the reasoning process level, they do not fully cover the area of TAMP to facilitate the planning process. For example, in motion planning, to deal with rigid bodies in cluttered environments, there is the need to define the way to apply actions such as push/pull, requiring a rich semantic description to be fed to the planner, like the physics-based motion planner in [10]. In task planning, the robot needs to reason on: (a) the feasibility of an action at some instant of the manipulation planning process (according to object features, the the state of the object could change and be out of the robot capabilities, e.g., if the cup is empty the state is graspable and if full the state is pushable), (b) the selection of the placement where the robot must place the object, and (c) the current constraints. Moreover, reasoning about perception knowledge is required.

PMK covers these gaps in the modeling and reasoning process levels. In the former, by following the standardized concepts presented by SUMO and CORA. In the latter, by providing reasoning predicates to cover TAMP needs.

## 4. Knowledge Formulation

The classes mentioned above in Section 3.2 are shown in the metaontology layer in Figure 4. The concepts in these classes are formulated here according to the standardized concepts presented in the IEEE 1872 standard. The ontology schema and ontology instance layers are not discussed since they depend on specific domains (in the motivation example, these have been built for the domain of our robotic lab). The formulation according to the standard uses mainly SUMO, CORA, CORAX, and ROA concepts.

SUMO divides the entities into two groups: *physical* and *abstract*. The *physical* group describes the entities that exist in space-time, and is subdivided into *object* and *process* to represent, respectively, bodies and procedures. The *abstract* group describes the entities that do not exist in time and include mathematical constructs [11].

Following the same modeling strategy, as shown in Figure 5, PMK divides the knowledge into knowledge related to objects (manipulation world), knowledge related to processes (manipulation planning), and abstract knowledge (manipulation data). The manipulation world knowledge includes the description of objects, robots and sensors in the workspace. The manipulation planning includes the reasoning processes over PMK. These are detailed in the next subsections. The manipulation data knowledge represents the features of objects, such as color, mass, robot constraints (e.g., joint limits) and sensor constraints (e.g., maximum and minimum measurement ranges).

### 4.1. Manipulation Data Knowledge

It includes the *Feature* class, which is subdivided into three gradual levels to represent data from low-level to high-level (abstract). The standardized concept of SUMO: *Quantity* is used in level one to describe the quantities of the environment, such as object dimensions, color, position, or orientation. A new concept *Quantity Aggregation* has been introduced in level two to include aggregations of quantities, like wrench (force and torque) or pose (position and orientation). The standardized concept of SUMO: *Attribute* is used in level three to represent abstract data like matrices. These concepts are modeled and related to the ontology schema layer (which in our case describes the IOC-Lab domain) to retrieve the required information. For instance, as shown in Figure 6, the *Quantity* concept is related to the concept *Pose* to retrieve the information of the grasping pose of YuMi gripper.

### 4.2. Manipulation World Knowledge

It includes four classes *WSobject, Actor, Sensor*, and *Workspace*. The main level of the first three classes is level two. In these levels, the standardized concepts are used to define the physical objects and their functionalities, while level one (component-level) represents the description of the components of the objects and level-three (grouping-level) represents the description related to the grouping of the objects in the world. For example, a *cup* has a body and a handle as components, and it can be grouped with a *saucer*.

As shown in Figure 5, in the main level of *WSobject, Actor*, and *Sensor* classes, the standardized concepts of SUMO: *Artifact*, CORA: *Robot* and SUMO: *Measuring Device* have been used, respectively. Component-levels use the new concepts *Artifact Component, Robot Component* and *Measuring Device Component*. Grouping-level uses the standardized concepts of SUMO: *Collection* and CORA: *Robot Group* for the *WSobject* and *Actor* classes, respectively, and the new concept of *Measuring Device Group* for the *Sensor* class. These concepts are modeled and related to the IOC-Lab ontology schema layer to retrieve the required information. Figure 6 shows an example of the relation between the component-level of the robot and the physical robot, where the YuMi robot and its gripper are defined under the *Robot* and *Robot component* meta-concepts, respectively.

The main level of *Workspace* class uses the concept from CORAX: *Physical Environment* that describes the topology of the objects in the environment (i.e., which area is occupied by which physical object), the standardized concept SUMO: *Region* is used to describe the geometrical representation of the workspace in level-one, and the new concept of *Semantic Environment* is introduced in level-three to complete level-two with the data (features) of the physical objects.

### 4.3. Manipulation Planning Knowledge

The manipulation planning knowledge represents the part that is responsible for reasoning about the situation of the objects and the robot, and for planning the tasks. This reasoning process is done over the manipulation environment knowledge to facilitate the planning process of the tasks. As shown in Figure 5, this knowledge includes two PMK classes, *Context reasoning* and *Action*. Each class has three levels, from high-level to low-level concepts. Level-three represents the symbolic information that depends on the first two levels. For example, in the *Context Reasoning* class, the standardized concept SUMO: *Situation* represents the status of the object or robot in the world with respect to space and time. To cover both, *Spatial Context* (such as *left, right, on, in*) and *Temporal Context* (such as *before, after, meet, overlap*) are introduced to describe them, respectively. These concepts are modeled and related to the IOC-Lab ontology schema layer to retrieve the required spatial information about the environment entities. For instance, Figure 6 shows the *can* object, which is sub-class of *Artifact*, that has the property *Spatially Located* to report the ontology with its spatial location with respect to other objects, e.g., the can is *on* the small table.

For *Action* class, symbolic tasks such as *serve* are defined in level-three (*Task*), which are composed of short-term sequences of simpler actions such as *move, pick up, moveholding, place*, that are defined in level-two (*Sub-Task*). The standardized concepts of ROA: *task* and ROA: *subtasks* are used to describe these two levels. Level-one includes the new concept of *atomic function* to represent processes for motion, manipulation, and perception, such as task planners, motion planners, or perceptual algorithms. Moreover, it includes primitive actions, preconditions, and postconditions related to manipulation behaviors. Although ROA provides the concept of *robot behavior* [20], it does not fully cover the manipulation planning perspective that we need. On the one hand, for sensors, the corresponding suitable algorithms (depending the type of sensor) should be available to extract the features of the environment (see Section 4.4). On the other hand, for planning (e.g., task and motion), the corresponding algorithms should be available to plan according to the problem. So, the new concept of *Atomic Function* is introduced to define the algorithms related to planning in terms of motion and task, and the perceptual algorithms.

### 4.4. Knowledge Representation for Perception

To perceive a robot environment, different sensors are usually used. Sensors provide data about the environment in the form of signals (one dimension) or images (multi-dimension), and to obtain the useful features from the perceived data the suitable algorithms have to be applied, for instance to detect an object pose some pose estimation algorithms based on image features can be applied, or alternatively algorithms based on tags identification can be used.

In Figure 5, sensors are defined in sensor class as measuring devices (level-two). The sensors, which can be attached with the robot or fixed in the environment, may contain different parts, which are represented as sensor components (level-one), like the tags, antenna, and reader that contain a RFID sensor. Several sensors may also be grouped as a device group (level-three) for a given application or domain, like the grouping of a RFID sensor and a 2D camera for object localization. This grouping, for instance, allows the robot to understand that it has two types of sensors to locate objects, as well as their differences, e.g., type of data extracted from each sensor, algorithms to be used on the data provided, or the best environmental conditions for their use. The sensor grouping concept makes the robot aware of its available equivalent sensing strategies and allow the selection of the proper sensor to use in each case according to the situation. PMK provides the relation between the *Feature, Sensor*, and *Action* classes that allows to extract the information from sensors. For instance, as shown in Figure 6, the concepts of *Camera* and *Algorithm*, which are inherited form *Measuring Device* and *Atomic Function* meta-concepts, respectively, are modeled and related to the IOC-Lab ontology schema layer to retrieve the required algorithm to extract the image features.

The inference mechanism required for the sensing procedure is detailed in Section 5.4. The main advantage of this way of representation is that it is workable for any type of sensor and any type of data processing algorithm (like pose estimation from tag detection in 2D images implemented in the ar–track–alvar library and used in the case study shown below).

## 5. Case Study

### 5.1. Task Description

Consider a manipulation problem including the bi-manual YuMi robot and a set of objects as depicted in Figure 1. The task is to *serve* the *wineglass* on the *servingTable*. Initially the *wineglass* is located inside the *box*. Due to reachability limitations, both arms have to collaborate with each other to solve the task. The responsibility of YuMi right arm is to pick up the *wineglass* and place it on the *smallTable* and then, using the left arm, pick up the *wineglass* up and place it on the *servingTable*. The challenge is that the placement region of the *wineglass* on the *smallTable* is already occupied by the *can*. Any planner to be used to solve this task needs a rich semantic description of the scene, able to answer questions such as *what are the sensors the robot has?*, *what are these sensors detecting?*, *what is the associated algorithm to extract the object features?*, *what is the spatial situation of the objects?*, *what is the available regions to place the object?*, or *how can the obstacle be removed?* and some other questions highlighted in Figure 7. PMK can provide answers to these kind of questions.

The perception module consists of two 2D cameras and tags for all the objects (see Figure 1). One camera is fixed on the top of the main table to sense the main table entities, and the other is attached to the YuMi left arm to perceive the serving table. The tags are used to identify the world entities and semantically link them to the the properties of each object. Specifically, the purpose of the perception module is to detect the position of the objects and their IDs and assert them on the ontology to build the ontology schema and the ontology instance layers of the IOC-Lab environment (as shown in Figure 3).

An expressive inference process helps to identify the hidden knowledge and increase the robots capabilities. The PMK framework uses an inference mechanism that consists of Prolog predicates. These predicates are used to query over the ontology to obtain the knowledge that the robot requires to manipulate the objects in the environment. The inference mechanism for manipulation planning domain includes the reasoning process related to sensing, task planning, and motion planning. The related generic predicates are explained in the following subsections.

### 5.2. Implementation

PMK can be integrated with any task and motion planner such as [26] to compute the sequence of actions to solve a manipulation task. The planner may ask the ontology questions about *how to perform the actions, what are the objects’ poses*, or *which are the interaction parameters of the objects?*. The PMK handles the requests of the planner and answers by retrieving information, updating/deleting or reasoning over it. The request-answer relation is done using the service-client communication of ROS (Robot Operating System, www.ros.org). The PMK is designed using ontology web language (OWL) with Protégé ontology editor (http://protege.stanford.edu/). Ontology instances can be asserted using information processed from low-level sensory data. The C++ library *ar_track_alvar* (http://wiki.ros.org/ar_track_alvar) has been used to detect the object pose and ID. These data are asserted in the PMK to extract a semantic description of the object. The 2D cameras have been used as a measuring device with two ROS nodes called *FixedCam* and *AttachedCam*, for the fixed and attached cameras, respectively. All the transformations of the objects and camera are calculated with respect to the world frame located at the YuMi base. Queries over the PMK are based on SWI-Prolog and its Semantic Web library which serves for loading and accessing ontologies represented in the OWL using Prolog predicates.

### 5.3. PMK Ontology Representation

In the metaontology layer, *region* is a concept linked to *artifact* and *quantity*, as shown in Figure 8 where the relation between the ontological layers that describe the IOC-Lab under the metaontology concepts is illustrated. In the ontology schema layer, the IOC-Lab region has artifacts such as *box, table, wineglass, can, cup* and quantities such as *color* and *dimension*. In the ontology instance layer, the instance *box01* of the subclass *box* represents the storage area that contains the instance *wineglass01* and *cup01* of the subclasses *wineglass* and *cup*, respectively. These instances have perceptual properties such as pose and tagID, that are asserted with the following predicate:rdf_assert(instance,registerName:objectproperties,assertedvalue), and linked to the fixed properties stored on the ontology, so as to have a complete knowledge about the objects. Then, once the the robot figures out the object ID, all the features can be extracted.

### 5.4. Reasoning Process on Perception

The inference process related to perception knowledge is basically the reasoning about feature extraction algorithms for perception, such as *FixedCam* or *AttachedCam* nodes used to extract the poses and IDs from images. The inference process related to the perception knowledge of PMK basically depends on the relation between three classes (*Feature, Sensor*, and *Action*). As shown in Figure 9, the *device* is a concept that has *quantity* and *atomic function*. The *quantity* contains the constraints and perceptual data. The constraints describe the sensor limitations, such as the fixed camera measuring only the *mainTable* with certain minimum and maximum ranges. The perceptual data describes the type of data and its properties, e.g., pose and IDs are the properties that are extracted from an image. These data are divided into two main parts *Feature of Interest* and *Observable Property* (names are inspired from the Semantic Sensor Network Ontolgy, https://www.w3.org/TR/vocab-ssn/). The former describes the type of data that a sensor senses, such as camera senses images. The latter is an observable characteristic (property) of a *Feature of Interest*, such as color is one of the image properties. As an example, the *RunFixedNode* predicate, shown below using DL, is applied to reason about the location and ID of the objects by being associated with the *FixedCam Node* atomic function (algorithm).

$$\begin{array}{c} RunFixedNode:\text{-}\\ \exists hasSuperclass(subTask,Action)\\ \wedge \exists \mathrm{Is}\text{-}a(FindObject,subTask)\\ \wedge \exists \mathrm{Is}\text{-}a(FixedCam,measuringDevice)\\ \wedge \exists hasfeatureOfInterset(2DCamera,Image)\\ \wedge \exists hasObseredProperty(Image,location)\\ \wedge \exists hasObseredProperty(Image,ID)\\ \wedge \exists useProcess(FixedCam,FixedNode) \end{array}$$

### 5.5. Reasoning Process on Situation

Now, the question that arise is, *what is the situation of the world entities?*. An evaluation-based process inspired from [13] has been introduced to spatially analyze the situation of the environment (e.g., *box on mainTable, wineglass01 inside box, smallTable left box)*. As shown in Figure 10, the spatial relations *right, left, on*, and *inside* have been introduced with some conditions, as detailed in Table 1, based on the bounding boxes of the objects in the environment. For example, to make the robot capable to understand that *box01* is *on* the *mainTable*, the predicate must check if the XY-projection of the bounding box of the top artifact lies within that of the bottom one, and then check for the Z-coordinates.

After the situation analysis is done, the PMK can provide useful information regarding the capabilities of the robot, placement regions and action constraints. For example, as shown in Figure 1, the tagged placement region of the *wineglass* is the same one that the *can* is occupying. The query to the knowledge to reason about the placement region is:$$\begin{array}{c} OccupiedPlacementRegion:-\\ \exists \mathrm{hasregionID}(\mathrm{obj},\mathrm{ID})\\ \wedge \exists \mathrm{hasspatialRelation}(\mathrm{obj},\mathrm{on}). \end{array}$$

### 5.6. Reasoning Process on Discrete Actions

To place the *wineglass*, first the placement region should be cleared from the obstacle *can*. For that, PMK must reason about the robot capabilities with respect to the objects to be manipulated, in order to determine which actions should be involved. The robot has a maximum payload and a given aperture of the gripper, and the objects contain information regarding the parts from where they should be grasped, as well the regions from where they should be pushed (e.g., a region defined around the object and below its center of mass). Then, for instance, to evaluate the feasibility of the pick up action, the size of the part to be grasped with respect to the gripper aperture has to be verified as well as the weight of the object with respect to the robot maximum payload. In this example the YuMi robot is able to pick up the *wineglass* but not the can (due to both its size and weight), which therefore needs to be pushed in order to be removed from the placement region it is occupying. The following DL predicate illustrates the reasoning about the capability of *pick up* action:$$\begin{array}{c} Pickup:-\\ \exists hasSuperclass(Pick,Action)\\ \wedge \forall hasTaskTarget(Pick,Artifact)\\ \wedge \exists hasArm(Robot,Arm)\\ \wedge \exists hasConstrains(Artifact,Top)\\ \wedge \exists hasGaspingPose(Artifact,GraspingPose)\\ \wedge \exists hasObjectPose(Artifact,ObjectPose)\\ \wedge \exists hasdimension(Artifact,Bbox_{1})\\ \wedge \exists isMemberOf(Gripper,Robot)\\ \wedge \exists hasdimension(Gripper,Bbox_{2})\\ \wedge \exists fitInside(Bbox_{1},Bbox_{2})\\ \wedge \exists hasCapability(Payload,Gripper) \end{array}$$

### 5.7. Reasoning Process on Motions

Sometimes in cluttered environments no collision-free motions exist to move the robot arm to a grasping pose, although a path toward the goal can be found if interactions with movable obstacles are allowed (i.e., the robot clears the path by pushing the obstacles away). This is done using physics-based motion planning strategies such as [10]. Briefly, physics-based motion planning is the evolved form of kinodynamic motion planning, that while planning do not preclude collisions with some obstacles and considers both kinodynamic constraints (such as joint limits and bound over velocities and forces) and physics-based constraints (such as friction and gravity).

The inference process for physics-based motion planning queries over the knowledge and reasons about the manipulation constraints. These involve the manipulation regions (regions around the object from where the robot is allowed to interact with the objects), and also the interaction dynamics parameters like friction and damping coefficients, and bounce velocities defined for the physics engines. A DL predicate to query to the knowledge to reason about physics-based manipulation constraints is the following:$$\begin{array}{c} ManipulationConstraints:-\\ \exists hasSuperclass(artifact,WSobject)\\ \wedge \exists hasmaterial(material,artifact)\\ \wedge \exists hasmass(mass,artifact)\\ \wedge \exists hasInterParameter(interactParameter,artifact)\\ \wedge \exists hasManReg(manipulatableRegion,artifact) \end{array}$$

### 5.8. Extended Spatial Reasoning

For grasping and placing the *wineglass*, it is necessary to select the arm to execute the action and the placement region. For example, the right arm is selected for picking up the *wineglass* from the *box*, since it can not be done by the left one due to kinematic limitations. Moreover, to place it in a *smallTable* or *servingTable*, suitable regions are defined. To automatically reason about the reachability, the reachability space is represented as follow.

*Reachability Space Representation*: As shown in Figure 11, the three regions have been introduced; right, left, and middle, and a given set of labels per region indicate where objects can be placed. The right arm is responsible for tackling the objects in the right area, while the left arm is responsible for grappling with the objects in the left area. Both arms can be used to handle the objects in the middle area. The predicate *robot-reachability-grasping(Artifact, ?reachableArm)* has been used to figure out which arm can be used to grasp an object, while *robot-reachability-placement(Arm, ?PlacementTargetRgn)* has been used to figure out the target placement regions to place an object.

As an example, consider a *cup* is placed at *PR1* (Figure 11). Then, the predicate *robot-reachability- grasping(Artifact, ?reachableArm)* evaluates to true for the right arm of the YuMi, and false for the left arm. Then the predicate *robot-reachability-placement(Arm, ?PlacementTargetRgn)* can be used to verify that the right arm can place the cup at *PR1*, *PR2*, *PR3*, or *PR4*.

### 5.9. Task Planning and Execution using PMK

This subsection describes the usage of PMK predicates to execute the manipulation task. Figure 10 shows the ontology layers regarding the task planning concepts. In the metaontology layer, the relation between *task*, *subTask*, and *atomic function* is shown. In the ontology schema and ontology instance, related to the current example, it is shown that the robot has the ability to *serve* wine or a soft drink by defining two instances *serveWine* and *serveCan* (i.e., pick up the *wineglass* or *cup* from the *box* to start filling them for the customers regarding their demand *serveWine* or *serveCan* respectively). The task is divided into a sequence of Sub-task, while the actual actions and their behaviors are described as atomic functions. In this sense, *Atomic Function* have quantities to define constraints (e.g., which part of the object can be grasped?).

The primary requirement is to run the corresponding algorithm to each camera to estimate the objects position according to the environment reference frame, as shown in Figure 7. The predicate *RunFixedNode(FixedNode,? Algorithm)* is running as discussed in Section 5.4. The result of the query is *FixedCamNode* (as shown in Figure 12b) that recognizes images and extracts the pose and ID of the *mainTable* entities. Then, to set the initial scene of the environment to the planner, the poses are used to spatially analyze the situation of each object using the proposed spatial predicates (e.g., *wineglass* inside *box*). The robot query to ask about which arm can be used to pick up the *wineglass*, and where the *wineglass* can be placed?, using *robot-reachability-grasping(Artifact, ?reachableArm)* and *placementRegion (Artifact, ?Region)*. Due to the occupancy of the tagged placement region of the *wineglass* by the obstacle *can*, the robot queries about the feasibility of available actions to remove the can, using the predicate *feasible(Artifact, ?Action)*. The results of these queries show that the object is *graspable* or *pushable*, according to the payload of the object and the capability of the robot. To know how to interact with the objects, the physics-based motion planning needs the interaction parameters and information from where the robot can interact (i.e., manipulation region). For this, the robot uses the *manipulationConstrains(Artifact, ?Properties)* predicate. The sequence of executed actions using these aforementioned predicates is shown in Figure 12c–f. Then, as shown in Figure 12g,h, the robot picks up the *wineglass* with the right arm and places it at the *servingTable*, according to the results of the predicates *robot-reachability-grasping(Artifact, ?reachableArm)* and *robot-reachability-placement(Arm, ?PlacementTargetRgn)*.

Figure 12 has shown the sequence of actions that have been executed in a real experiment, once the planning has been efficiently done with the help of PMK. The second part of the case study (not discussed in this paper) involves the TIAGo robot, responsible for presenting the beverage on the *servingTable* to the customers.

### 5.10. System Flexibility

To illustrate the capacity to adapt to similar environments that PMK gives to the robot, let us consider the case where the task is to serve a cup of coffee (instead of the wineglass) also located inside the box, and that the region on the small table is currently occupied by a wineglass (instead of by a can), i.e., different objects and at different locations are considered for the same *serve* task (Figure 13). The PMK reasoning capabilities allow the robot to handle the differences: (a) now the only feasible grasping configuration for the *cup* is from the top, instead of the lateral grasping configuration used for the wineglass; (b) to remove the wineglass from the intermediate location, a pick up and place action has to be done instead of the push action formerly done for the can.

Like done in the first example, first, the *RunFixedNode(FixedNode,? Algorithm)* predicate is called to run the corresponding algorithm to extract the features and evaluate the spatial relations between the entities, then *robot-reachability-grasping(Artifact, ?reachableArm)* and *placementRegion(Artifact, ?Region)* are applied to know which arm can be used and where the *cup* can be placed. The predicate *feasible(Artifact, ?Action)* can infer that the *wineglass* (which occludes the placement region of the *cup*) is within the robot capability to apply the pick up and place actions to remove it, as shown in Figure 13b–d. Then, to know the feasible grasp for the cup, *manipulationConstrains(Artifact, ?Properties)* is used. The sequence of executed actions using the aforementioned predicates are shown in Figure 13e–g. Finally, the robot can apply the *pick up* action using the *robot-reachability-grasping(Artifact, ?reachableArm)* predicate to know which arm is reachable, then *robot-reachability-placement(Arm, ?PlacementTargetRgn)* predicate to know where is the placement region to place the *cup* in the *servingTable*.

Therefore, provided that the objects to be manipulated are from the domain included in the ontology schema, the robot is able to adapt to new situations and accomplish the task by means of the perception and reasoning capabilities given by PMK.

## 6. Discussion

### 6.1. Discussion about the Results

The motivation example highlights the significance of using PMK to make the robot understand the current scenario and act according to the actual manipulation constraints. The manipulation constraints related to task execution can be listed as:Manipulation constraints such as: From where the object can be interacted?, What are the interaction parameters?Geometric constraints such as: What is the spatial robot reachability? Where can the objects be placed?Action constraints such as: From where can the actions be applied?Perception reasoning such as: What is the sensor attached to the robot? How does it work? What are its constraints, such as the sensor range measurement?

PMK provides the answers needed by combined task-motion planners and by physics-based motion planners.

Some other frameworks proposed a knowledge-based processing and reasoning such as [6,13]. Although there are some similarities with them in terms of spatial reasoning or reasoning about the object features which could be common in the working domains, the reasoning process related to planning (including geometric reasoning) and perception reasoning are newly proposed in PMK. For instance, these frameworks proposed some spatial relations such as *on, inside, right, left* for the robot to spatially related between the environment entities. The extended spatial reasoning that includes some extra predicate such as spatial robot reachability and arm selection for bi-manual robots have been introduced. In the case study, the importance of these predicates appear, for instance, when the robot needs to execute the pick up action for *wineglass*. First, by asking about which arm is reachable, then by asking about the reachable placement region to place the *wineglass* on the *servingTable* with the left arm. Moreover, when using a physics-based motion planning, to apply the *push* action, a query is posed to know how to interact with the obstacle *can*, which needs, for instance, the interaction parameters like the friction coefficient.

Moreover, for perception, PMK proposes the reasoning related to the perceptual features of the objects in the environment, like others, but also some reasoning related to the suitable algorithms that the sensor can run to extract the features, the sensors that are associated with the robot, and the sensor features and its limitations. The perception reasoning process makes the robot more smart and flexible. This flexibility can be useful to cope with the failure of a sensor, by providing an alternative one, i.e., PMK has the flexibility to deal with multi-model sensory systems.

### 6.2. Discussion about the System

The execution of manipulation tasks with knowledge-based planning approaches not explicitly prepared for TAMP, like [13], can be a challenge because this would require, on the one hand, from the knowledge perspective, to provide all the components in a way that they match with their planning system. On the other hand, from the planning perspective, they would require the definition of the recipes (strategies) for executing the tasks (sequence of actions), including all possible strategies of execution and the way to switch between them when required, which can be a very expensive process, especially for tasks that need long sequences of actions, such as those involving manipulation in cluttered environments. Some other planning approaches rely on PDDL and on planning strategies best fit to cope with difficult task planning challenges, like those found in the manipulation domains, although the use of PDDL implies a closed-world assumption, which precludes their use in more dynamic environments that could require perception and knowledge-based geometric reasoning.

The main role of PMK is to facilitate the planning process for TAMP by providing the necessary planning components, such as geometric reasoning, dynamic interactions, manipulation constraints, and action constraints. Moreover, it includes the perception knowledge and reasoning about the sensors that the robot has, the corresponding algorithms to extract the features, and the sensors limitations. All these components are required to automatically execute complex tasks, like those presented in [4,27], and to make the system flexible enough to adapt to different situations requiring different manipulation actions, as demonstrated with the two motivation examples.

## 7. Conclusions and Future Work

This study proposes the formalization and implementation of the standardized ontology framework PMK to extend the capabilities of autonomous robots related to manipulation tasks that require task and motion planning (TAMP). In terms of modeling, the aim has been to contribute with a unified framework based on standardization that provides common vocabularies in order to have the flexibility to incorporate PMK with other ontologies, to avoid conflict in the meanings of concepts. Moreover, in terms of reasoning process, some important components for task, motion, or combined task-motion planning are proposed, such as reasoning for perception, reasoning about the object features, reasoning about the environment, geometric reasoning, and reasoning for planning.

To illustrate the proposal, an example has been introduced to show the PMK framework abilities to query knowledge about the geometric reasoning and robot capabilities to solve TAMP problems. Most of the requirements for TAMP, discussed in the paper, are met by the proposed framework. The knowledge is organized in a way to facilitate the planning process, so that the robot can easily access the concepts it needs for its tasks, i.e., PMK enables the robot to complete a manipulation task autonomously in spite of hidden or partial data. To substantiate the perceived information, knowledge related to perception has been considered, which allows analyzing the situation of the environment. Since TAMP queries the requirements to PMK, it automatically adapts to different scenarios.

The PMK framework is currently being extended with more perception measuring devices, such as RFID sensors and 3D cameras, to be applied to more complex manipulation tasks involving assemblies. Also, an extension is planned to include concepts and reasoning related to failures, in order to make PMK useful in situations where the task planner may not be able to find a feasible sequence of actions to perform a given task, or may need to recover from an error.

## Figures and Tables

**Figure 1 sensors-19-01166-f001:**
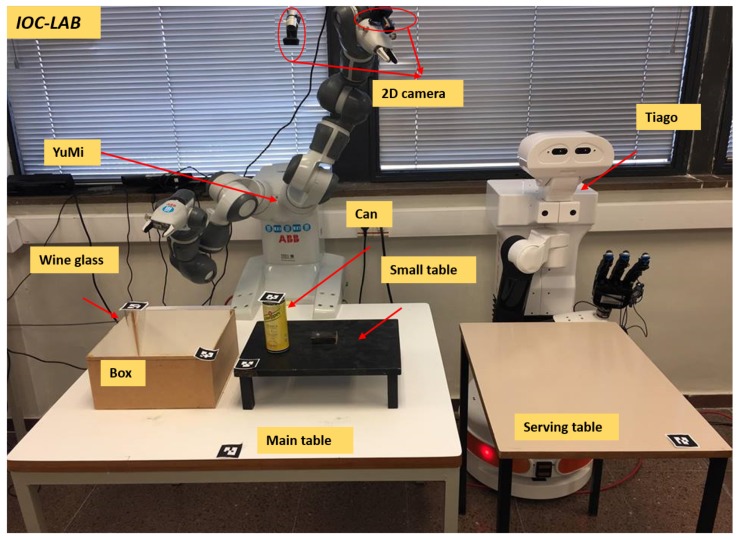
A motivation example of a two-robot table-top manipulation task at the IOC-Lab.

**Figure 2 sensors-19-01166-f002:**
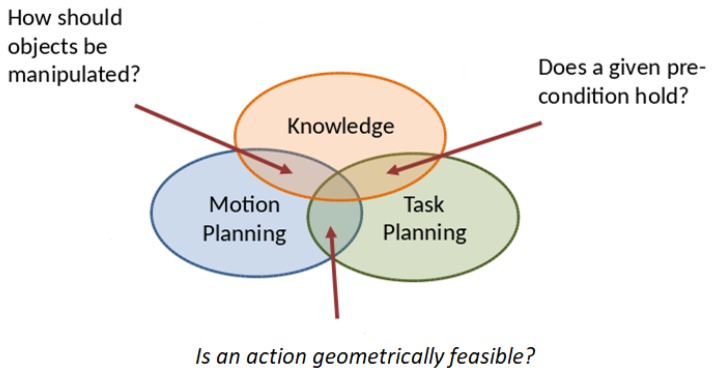
Knowledge-based reasoning for task and motion planning (TAMP).

**Figure 3 sensors-19-01166-f003:**
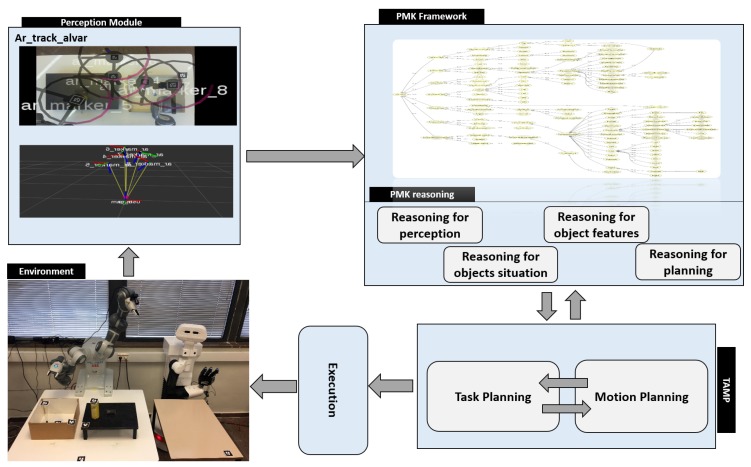
Main parts of the system: The perception module, the Perception and Manipulation Knowledge (PMK) framework, and the TAMP planning module. PMK asserts the perceptual data, builds the IOC-Lab knowledge, and provides the reasoning predicates to the planning module.

**Figure 4 sensors-19-01166-f004:**
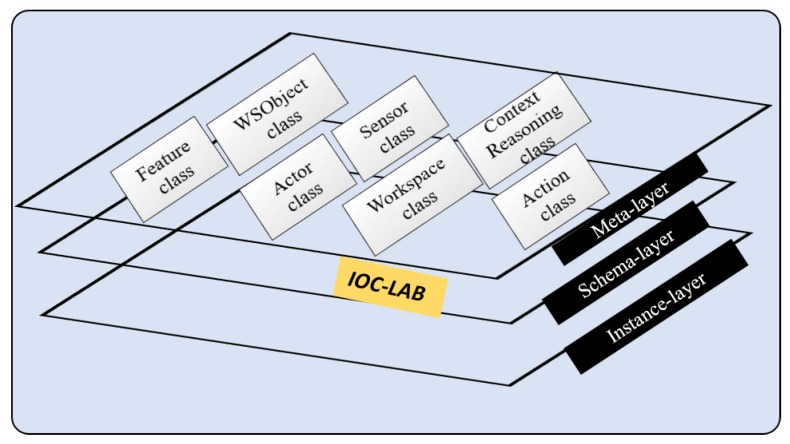
PMK ontology.

**Figure 5 sensors-19-01166-f005:**
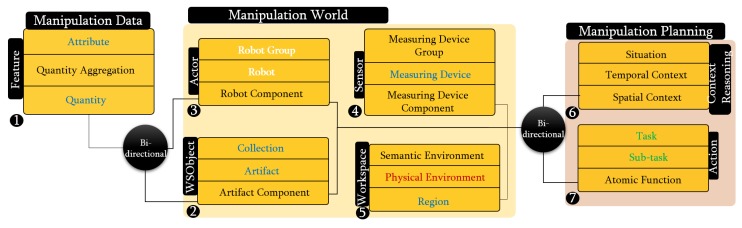
Structure of the metaontology layer of PMK fit to follow the standard IEEE-1872. Concepts introduced in this paper are shown in black, while those that inherit from the standard are shown in color: SUMO (**blue**), CORA (**white**), CORAX (**red**), and ROA (**green**). All the terminologies are defined in the Appendix A.

**Figure 6 sensors-19-01166-f006:**
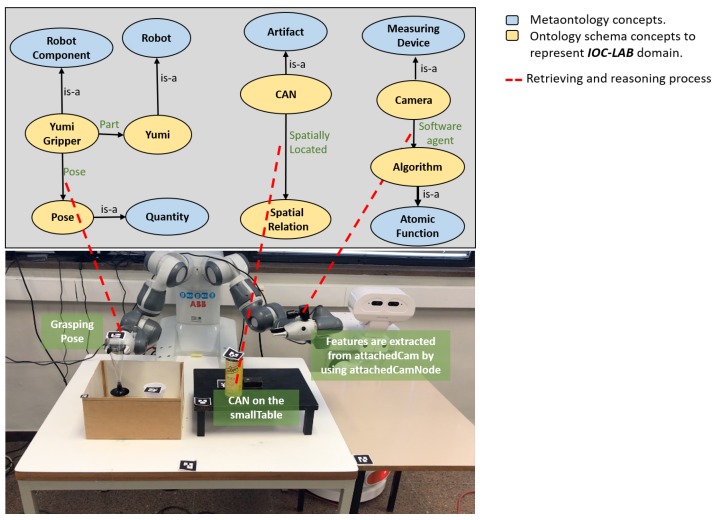
Description of some abstract manipulation knowledge concepts.

**Figure 7 sensors-19-01166-f007:**
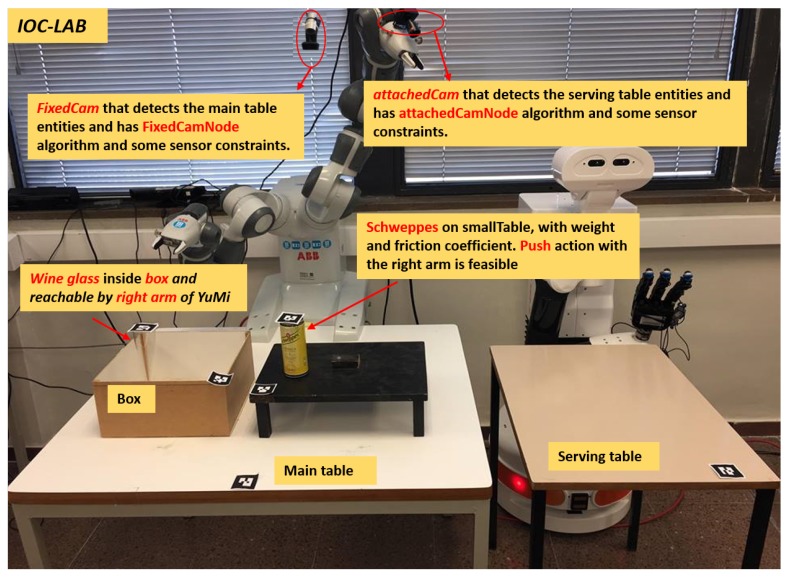
The description of the constrains of the motivation example related to sensors, object geometry, interaction learning, and action feasibility.

**Figure 8 sensors-19-01166-f008:**
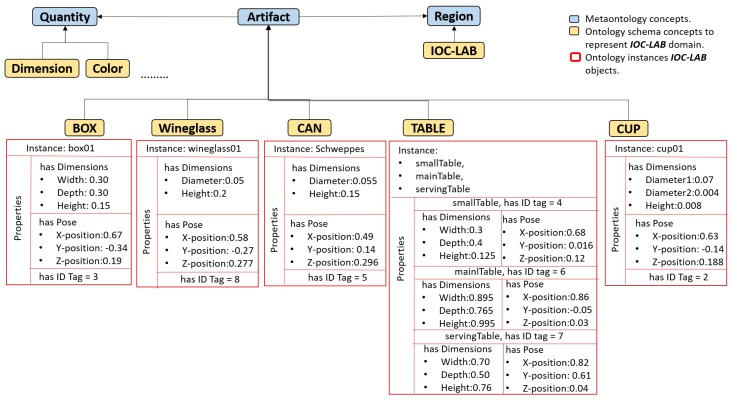
Taxonomy representation of the IOC-Lab domain in the PMK framework. The concepts of region, artifact, and quantity are inherited from Workspace, Wsobject, and Feature classes, respectively. The Quantity describes the features of each instance as shown in the properties part in the instantiated artifacts (the orientation information of the objects poses is not included in order to simplify the figure).

**Figure 9 sensors-19-01166-f009:**
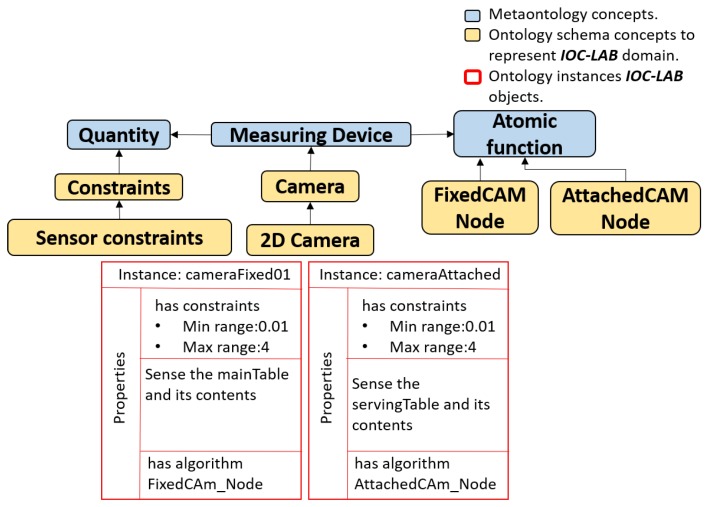
Taxonomy representation of sensor class and its relation in PMK framework. The Quantity and Atomic Function describe the sensor constraints and the labeled running algorithm of each sensor, as shown in the properties part, and inherit from Feature and Action classes, respectively.

**Figure 10 sensors-19-01166-f010:**
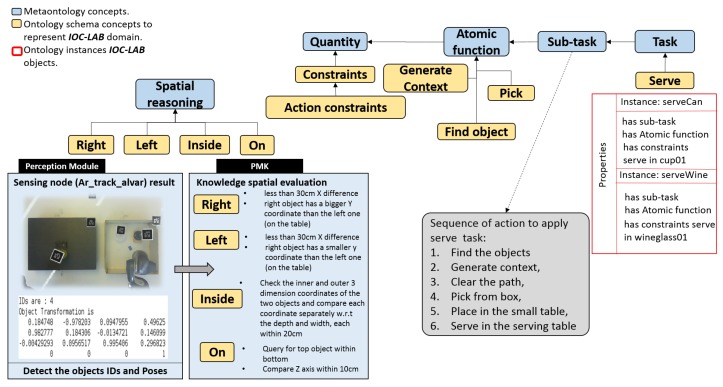
Schema of the flow of information between classes that belong to the metaontology, ontology schema, and ontology instance layers of manipulation planning classes. On the left, the flow of asserted data from perceptual module and the spatial evaluation process. On the right, the task and the sequence of actions needed to execute it.

**Figure 11 sensors-19-01166-f011:**
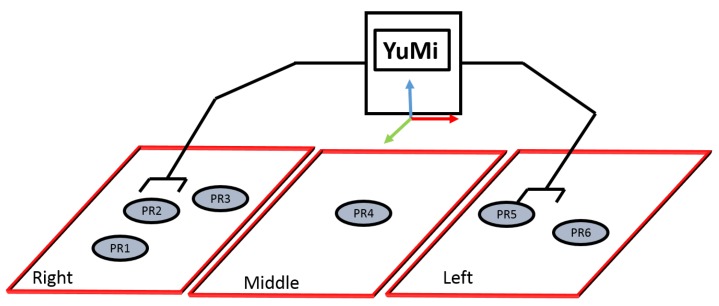
The reachability space representation.

**Figure 12 sensors-19-01166-f012:**
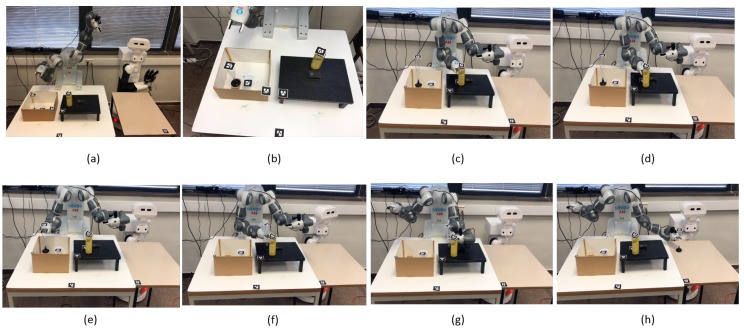
The sequence of snapshots for the execution of the *serve* a glass of wine instruction. Video URL-link “https://sir.upc.edu/projects/kautham/videos/PMK-final.mp4”.

**Figure 13 sensors-19-01166-f013:**
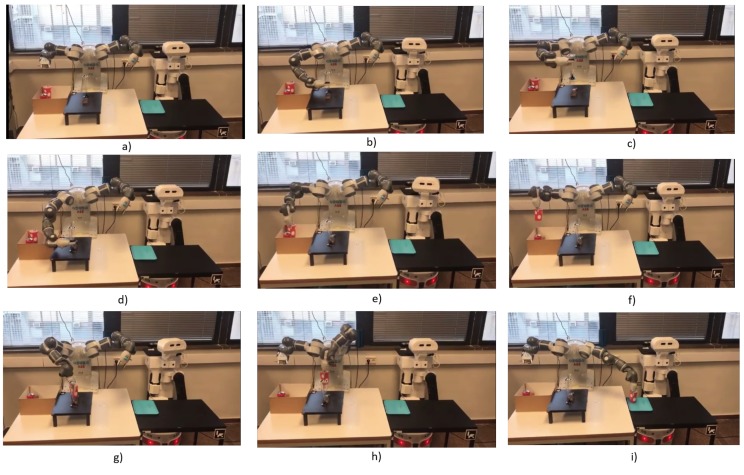
The sequence of snapshots for the execution of the *serve* a cup of coffee command. Video URL-link “https://sir.upc.edu/projects/kautham/videos/PMK-final.mp4”.

**Table 1 sensors-19-01166-t001:** Spatial reasoning.

DL Description	Condition	Effect
*(On, Subjective, Objective)* :- ∃hasSuperclass(Spatialcontext, Contextreasoning), ∧∃hasSubjective(On,artifact01), ∧∃hasObjective(On,artifact02).	The X and Y ranges of the bounding box of the Subjective artifact must be within those of the Objective, and the Z-ranges must be contiguous.	t artifact01 *On* artifact02
*(inside, Subjective, Objective)* :- ∃hasSuperclass(Spatialcontext, Contextreasoning), ∧∃hasSubjective(inside,artifact01), ∧∃hasObjective(inside,artifact02).	The X and Y ranges of the bounding box of the Subjective artifact must be within those of the Objective, and the Z-ranges must overlap.	t artifact01 *inside* artifact02
*(Right, Subjective, Objective)* :- ∃hasSuperclass(Spatialcontext, Contextreasoning), ∧∃hasSubjective(right,artifact01), ∧∃hasObjective(right,artifact02).	The X and Z ranges of the bounding box of the Subjective artifact must overlap those of the Objective, and the X-range of the bounding box of the Subjective must have its minimum value greater than the maximum value of the X-range of the Objective.	t artifact01 *right* artifact02
*(left, Subjective, Objective)* :- ∃hasSuperclass(Spatialcontext, Contextreasoning), ∧∃hasSubjective(left,artifact01), ∧∃hasObjective(left,artifact02).	The X and Z ranges of the bounding box of the Subjective artifact must overlap those of the Objective, and the X-range of the bounding box of the Subjective must have its maximum value lower than the minimum value of the X-range of the Objective.	artifact01 *left* artifact02

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
