# Peer review of "PMK—A Knowledge Processing Framework for Autonomous Robotics Perception and Manipulation"

_sensors, 2019, doi:10.3390/s19051166_

Reviewer 1 Report

The authors presented a perception and manipulation framework for objects manipulation. The frame work model the environment in order to extract necessary information for task driven actions. It also takes in consideration sensory information to assert ontological knowledge. And finally the planning scheme is based on the evaluation of the objects situation and feasible actions.

The reviewer has few remarks and suggestions to the authors:

- What does PMK stands for (in the abstract)?

- In 4.4. Knowledge representation for perception  sensors grouping is confusing, and overlap with each other.

- PMK does not provide semantics of the scene, AR tags does.

- Objects are placed in 3d space, thus they have 6Degrees Of Freedom (DOF), however, from figure 7 the description show only the linear pose of the objects. How about the angular configuration of objects in workspace? This is important for manipulation since the trajectory of the end effectors of both arms needs to be generated, and this cannot only be based on linear position of the objects to be grasped.

- The authors claims that the scheme is generic, however, a detailed description strictly attached to the objects present in the scene is given, which make the proposed solution looks like it working only for the subset of objects already described, can the authors elaborate more on this point?

- It can be beneficial to the paper to include other experiments with different configurations of the objects inside the scene to evaluate the reaction of the system.

- No discussion of the case that no reachable position is found, which can be the can in randomly placed objects inside the scene.

Author Response

Thanks for the reviewer for the fruitful revision. I have attached the answers to the reviewers with the manuscript in red.

Reviewer 2 Report

The authors presented a PMK framework proposed for objects manipulation. The presented work is undoubtedly an interesting solution in the field of robot orientation in the environment, extraction of environmental information and taking actions on their basis. Unfortunately, the material presented was described in a not very transparent way. The result is the presence of large and difficult to analyze blocks of text in the context of the few figures illustrating the ideas of the designed framework. On this basis the reviewer has a few suggestions for the authors.

1.      In the points 3.4, 4, 4.1, 4.2, 4.3, 4.4 and 5.1 proper figures would simplify the understanding of the contents of sub-chapters.

2.      It is no clear why there is no word about experiments with different configurations of the objects inside the scene. This would evaluate the reaction of the proposed system. It seems that Authors have resources to do such experiments so it should be clearly stated.

3.      There is no discussion about system reaction to random objects. For the reviewer it is clear that Authors have necessary equipment to do proper experiments so it would be beneficial for entire work to conduct and discuss them. 

Author Response

Thanks for the reviewer for the fruitful revision. I have attached the answers to the reviewers with the manuscript in red.

Round  2

Reviewer 2 Report

The presented explanations are sufficient. The reviewer thanks the Authors for their corrections and answers.